# An Integrative Evaluation Method for the Biological Safety of Down and Feather Materials

**DOI:** 10.3390/ijms20061434

**Published:** 2019-03-21

**Authors:** Toshikatsu Kawada, Junya Kuroyanagi, Fumiyoshi Okazaki, Mizuki Taniguchi, Hiroko Nakayama, Narumi Suda, Souta Abiko, Satoshi Kaneco, Norihiro Nishimura, Yasuhito Shimada

**Affiliations:** 1Kawada Feather Co., Ltd., Nagoya, Aichi 453-0064, Japan; toshi@kawada.net; 2UMOU Science Lab, Mastusaka, Mie 515-0074, Japan; kuro@umou.or.jp (J.K.); suda@umou.or.jp (N.S.); 3Department of Life Sciences, Graduate School of Bioresources, Mie University, Tsu, Mie 514-8507, Japan; okazaki@bio.mie-u.ac.jp (F.O.); 515557@m.mie-u.ac.jp (M.T.); 4Department of Bioinformatics, Mie University Advanced Science Research Promotion Center, Tsu, Mie 514-8507, Japan; 5Mie University Zebrafish Drug Screening Center, Tsu, Mie 514-8507, Japan; 27293301@m.mie-u.ac.jp (H.N.); nishimura.norihiro@mie-u.ac.jp (N.N.); 6Graduate School of Regional Innovation Studies, Mie University, Tsu, Mie 514-8507, Japan; 7Boken Quality Evaluation Institute, Minato-Ku, Osaka 552-0021, Japan; abikoa@boken.or.jp; 8Department of Chemistry for Materials, Graduate School of Engineering, Mie University, Tsu, Mie 514-8507, Japan; kaneco@chem.mie-u.ac.jp; 9Department of Integrative Pharmacology, Mie University Graduate School of Medicine, Tsu, Mie 514-8507, Japan

**Keywords:** allergy, microbiome, pollution, environment, clothes, bedding

## Abstract

Background: Down and feather materials have been commonly used and promoted as natural stuffing for warm clothing and bedding. These materials tend to become more allergenic as they become contaminated with microorganisms, in addition to being subjected to several kinds of chemical treatments. The biological or chemical contaminants in these materials pose a major risk to human health, to consumers and manufacturers alike. Here, we report the development of an integrative evaluation method for down and feather materials to assess bacterial contamination and in vivo toxicity. Methods: To assess bacterial contamination, we quantified 16S ribosomal RNA, performed culture tests, and established a conversion formula. To determine in vivo toxicity, we performed a zebrafish embryo toxicity testing (ZFET). Results: Washing the material appropriately decreases the actual number of bacteria in the down and feather samples; in addition, after washing, 16S rRNA sequencing revealed that the bacterial compositions were similar to those in rinse water. The ZFET results showed that even materials with low bacterial contamination showed high toxicity or high teratogenicity, probably because of the presence of unknown chemical additives. Conclusions: We established an integrative evaluation method for down and feather safety, based on bacterial contamination with in vivo toxicity testing.

## 1. Introduction

For centuries, humans worldwide have used down and feather for insulation, in outerwear and in duvets. Since down and feathers are generally collected from ducks and geese, there is a possibility that using these products, especially pillows, can trigger health problems, including allergies to bird feathers. However, a Finnish study has shown that true down and feather allergies are rare (less than 0.5% of adult patients with suspected allergic cutaneous or respiratory symptoms), and most issues are caused by dust mites and bacteria through long-term use [1]. Currently, despite the development of hypoallergenic down and feather materials, the number of patients suffering from adverse health effects caused by these products, including “Down Pillow Allergy Symptoms”, is increasing [2]. Factors contributing to the health concerns include insufficient washing of the sewed and sealed final product (outwear, duvet, and pillow) and storage and long-term use by consumers. In addition to the allergies, chemical treatments also pose a serious threat to the health of consumers. Two types of chemicals are used to treat down and feathers—one being the chemicals used to prepare antimicrobial-impregnated down and feather materials, and the other, a group of sticky chemicals that bind dust and fibers to down clusters, which increase the weight of the product, the so-called “glue down” [3]. Following the process of glue down, excess fiber and dust are found in the products, which may be harmful to human health. Because glue down looks very similar to regular down, it is very difficult to identify glue down by simply looking at the material. It may create confusion in the market and may lead to concerns over quality and increase the difficulty of quality testing.

Rodents are widely used to test the potential developmental toxicity of chemical substances. Although reliable data for extrapolating toxicant effects to humans are obtained through laboratory rodent studies, these are expensive, time consuming, and highly restricted by ethics and law. Since genes, receptors, and molecular processes are highly conserved across animal phyla, zebrafish (*Danio rerio*) have become an attractive model for chemical toxicology testing centered around the zebrafish embryo toxicity test (ZFET) and is an alternative to rodent models [4,5]. In fact, the European Chemicals Agency strongly recommends the use of the ZFET for an in vivo assessment of the safety of chemicals used in household commodities. In addition, since juvenile zebrafish exhibit a transparent body wall, relatively uncomplicated assessments on the internal organ toxicities can be made by using transgenic zebrafish with tissue-specific fluorescent protein (e.g., EGFP) expression. For example, we previously reported that copper oxide nanoparticles inhibited physiological angiogenesis using vascular-EGFP strains, *Tg(fli1a:EGFP)*/*nacre* zebrafish [6].

The current regulations on the use of down and feather materials in consumer goods need to be renewed to allow for world trade and mass production. In the current study, we developed a PCR-based detection protocol for bacterial contamination with protein measurement in down and feather samples. Toxicity of down and feather materials was evaluated using the ZFET protocol with some modification, including the use of a macrophage-specific EGFP expressing fish line.

## 2. Results and Discussion

### 2.1. qPCR Quantification for Contaminated Bacteria in Down and Feather Samples

We first evaluated the potential for bacterial contamination of down and feather materials using a standard agar plate as the conventional culture method. The culture test is conducted to accurately assess bacterial contamination mainly in food products. In the present study, colonies formed by the feather extracts were counted and a colony-forming unit (CFU) was determined. Typical images by scanning electron microscope (SEM) of unwashed and washed samples are depicted in Figure 1a. Unwashed samples contained debris, which was not observed in washed samples (red arrowheads). As a natural consequence, washed feathers exhibited a lower CFU than unwashed samples (*p* < 0.001, Figure 1b). The average CFU of unwashed feathers was approximately 5.8 × 10^5^ cfu/g, which was acceptable, considering the standard 1 × 10^6^ cfu/g as per the “Feather and down-Test methods-Determination of microbiological state (The European Standard EN 1884)”. Generally, down and feather materials are washed once at the site of harvesting, so our result proved that these samples were pre-washed at that moment. However, one of the samples was above 1 × 10^6^ cfu/g, which requires washing according to the feather and down test regulation. In addition, in the food regulations, the bacterial content should be lower than 1 × 10^5^ cfu/g, in which case most of the feathers used were not acceptable in our result. Since down and feather quilt are sometimes used for bedding or cushions, we insist that the CFU standards for their use should be similar, if not the same as those for food chains.

The bacterial plating (CFU calculation) is a reliable and reproducible method despite certain disadvantages. For instance, it is not well suited for a large number of samples, due to the complicated procedures with serial dilutions of test samples and is time consuming. Manual counting of bacteria can also lead to human error. It is crucial to note that colonies develop only from those organisms for which the culture conditions are suitable for growth, only less than 1% appear to be cultivable on different nutrient media as reported in a previous study [7]. To overcome this problem, many researchers try to perform qPCR-based bacterial quantifications with comparisons drawn based on CFU content [8,9]. We performed qPCR for the bacterial 16S rRNA gene (V3 region) using a primer set that corresponds to regions highly conserved among bacteria, which usually leads to the amplification of the genome of the contaminating bacterium [10]. As shown in Figure 1c, qPCR-based estimation of bacterial numbers in unwashed samples were about 94-times larger than those of the CFU, reflecting the speculation of the previous study about the existence of non-cultivable microorganisms [9]. Importantly, the washed feathers contained a total number of bacteria approximately 100 times lower than unwashed feathers, quite a similar ratio to the CFU result. We conclude that qPCR-based quantification of bacteria is a method that is sensitive and parallel to conventional CFU analysis albeit with a high throughput.

To determine the actual proliferation of bacteria in down and feather samples, we exposed unwashed and washed samples to *E. coli* expressing the mCherry fluorescent protein, as a model for bacterial contamination. As shown in Figure 1d, there was no visible fluorescent signal in washed down and feather, while unwashed samples exhibited a red fluorescence, especially in the branched area. These areas are the same places in which the debris was found (Figure 1a). These results indicated that suitable washing can remove nutrients for bacterial growth. In addition, the protein contents of the immersed water of these samples were comparable to the results of the bacterial contamination (Appendix A), suggesting that one of the nutrients for bacterial growth would be the residual proteins in the unwashed down and feather materials, especially located in the branched area.

### 2.2. Compositions of Contaminated Bacteria in Down and Feather Samples

To identify the composition of contaminated bacteria in down and feather samples, we performed 16S rRNA sequencing (full-length) of 3 samples from different countries (1, France; 2, Taiwan; 3, Poland) before (unwashed) and after washing (washed). Although down and feather samples were from different countries, there is a similarity in bacterial compositions of unwashed samples. Firmicutes is a major phylum at the phylum level (Appendix A), and Bacilli is a major class at the class level (Appendix A). After washing, all samples exhibited similar composition to the rinse water, which was dominated by Proteobacteria in phylum level with Beta- and Gamma-proteobacteria in class level. In all samples, Bacilli class in the Firmicutes phylum was reduced by washing, and the frequency of the Beta- and Gamma-proteobacteria class increased, similar to the rinse water (Figure 2a for phylum and Figure 2b for class level). The bacterial composition would be an appropriate criterion to rank the state of cleanliness of down and feather materials, which can be done by generating a ration between the quantities of Firmicutes and Proteobacteria. Of course, the bacterial contamination of washed samples was quite lower than those of unwashed samples (Figure 2c).

### 2.3. In Vivo Toxicity of Down and Feather Samples

For the bacterial contamination of down and feather materials, we can quickly evaluate them by the qPCR-based methods as described above. However, other poisonous or harmful molecules, such as toxic chemicals, cannot be detected by this protocol. Recently, some companies have added sticky chemicals referred to as a “glue” to increase the volume of down and feather by formation of dust clump, in addition to antibacterial and antifungal chemicals to protect against microorganisms for their long-term use. The SEM images of glue down is depicted in Figure 3a. Red arrowheads indicate glue randomly attached on the feathers in these samples. To evaluate the biological toxicity of the unwashed, washed and glue down samples, we performed the zebrafish embryo toxicity test (ZFET), according to the OECD guideline [11] with some modifications. There was no significant difference in survival rate between these experimental groups with relatively large variance after 72 h (Appendix A). However, glue down exhibited a variety of abnormal phenotypes, including unhatching at 72 hours-post-fertilization (hpf) (Figure 3b), while unwashed and glue down significantly (*p* < 0.01) increased the toxicity toward zebrafish (Figure 3c). In addition, glue down showed high toxicity, while the bacterial contamination of glue down was lower (*p* < 0.01) than that of unwashed samples (Figure 3d). In fact, the used breeding water of zebrafish was clean in the glue down group, consistently with low-bacteria contamination. This indicates that other chemicals, such as antibiotics or antifungals, were added to the glue down, in addition to the sticky chemicals. To further evaluate immunotoxicity of down and feather samples, we performed ZFET with macrophage-EGFP zebrafish, *Tg(mpeg1:EGFP)* [12]. According to the immunotoxicity studies using zebrafish [13], some unwashed and glue down samples significantly (*p* < 0.001) increased the numbers of macrophages (green signal) in the tail fin of zebrafish (Figure 3e,f), suggesting that these samples have a possibility of inducing inflammation in human usage.

The China Feather and Down Industrial Association previously identified silicon compounds as sticky chemicals in glue down (unpublished data). We hypothesized that silicon compounds would be major toxicants. We quantified the silicon (Si) contamination in unwashed, washed, and glue down. Glue down contained higher levels of Si than the other down materials (Appendix A). However, we could not detect Si in the washed water of all samples (less than 1 ppm), suggesting that there is still the possibility that there are other unknown chemicals used in down and feather materials. Identification of chemicals is an expensive, labor-intensive, and time-consuming procedure. Instead, ZFET would be an appropriate alternative technology to detect toxicity due to its high sensitivity, the relative ease at which it can be performed, and its cost-effectiveness. In addition, using EGFP-expressing transgenic zebrafish provides an additional parameter, which enables the detection of hidden toxicities, accompanied by a normal gross phenotype (Figure 3d).

## 3. Materials and Methods

### 3.1. Scanning Electron Microscopy of Down and Feather Samples

The down and feather samples were prepared by Kawada Feather Co. Ltd (Nagoya, Aichi, Japan) and UMOU Science Lab (Matsusaka, Mie, Japan). The microstructures of the down and feather samples were imaged using TM-1000 scanning electric microscopy (Hitachi, Tokyo, Japan).

### 3.2. Zebrafish

Zebrafish AB and *Tg(mpeg1:EGFP)* strains (the Zebrafish International Research Centre, Eugene, OR, USA) were maintained in our facility according to standard operational guidelines. All animal procedures were approved by the Ethics Committee of Mie University, were performed according to the Japanese animal welfare regulation ‘Act on Welfare and Management of Animals’ (Ministry of Environment of Japan), and complied with international guidelines. Ethical approval from the local Institutional Animal Care and Use Committee was not sought, since this law does not mandate the protection of fish.

### 3.3. Bacterial Examination

The down and feather samples were examined for bacterial contamination using a standard agar plate (casein-peptone dextrose yeast) method. One gram of down and feather samples was immersed in 10 mL of 0.1% sterile peptonic physiological solution and incubated at 25 °C for 3 h. Then, 1 mL of serially diluted samples were plated, incubated at 30 °C for 48 h, and the number of colonies were counted, according to the European Standard EN 1884.

### 3.4. Escherichia coli Exposure

To create mCherry protein expressing *E. coli*, we transfected a pFCcGi plasmid (Addgene, Cambridge, MA, USA) to the DH5α bacteria with a standard protocol. We selected and established stable mCherry expressing cells (DH5α-mCherry), using ampicillin (100 μg/mL) selection. 0.1 g of down and feather samples were co-incubated with 1 mL of DH5α-mCherry culture at 37 °C overnight. Images were captured using a BZX-710 fluorescent microscope (Keyence, Tokyo, Japan).

### 3.5. qPCR

An InstaGene Matrix (Bio-Rad Laboratories, Richmond, CA, USA) was used to prepare the DNA template, according to the manufacturer’s protocol with some modifications. In brief, down and feather (0.1 g) samples were immersed in 10 mL sterile water and shaken for 1 min at room temperature. Then, 1 mL of the immersion water was collected from the 10 mL samples, and the down and feathers were removed by centrifugation (13,000× *g*, 1 min). The supernatants were added to 100 μL InstaGene Matrix to prepare for the DNA template. qPCR was performed using Power SYBR Green Master Mix (Applied Biosystems, Foster City, CA, USA) and the ABI StepOnePlus Real-Time PCR System (Applied Biosystems), in accordance with the manufacturer’s instructions. For oligonucleotide primers, 16S-341F (5′-CCT ACG GGA GGC AGC AG-3′) and 16S-518R (5′-WTT ACC GCG GCT GCT GG-3′) [14] were used to amplify the 16S rRNA-sequence. To obtain the standard curve, we incubated the down and feather (0.01 g) samples in 10 mL LB medium at 37 °C overnight, following which the medium was collected, and the number of bacteria were counted manually under a microscope. A DNA template was prepared using the InstaGene Matrix method as described above.

### 3.6. 16S rRNA Sequencing

In order to isolate DNA, down and feather (50 mg) samples were homogenized in 750 µL of Lysis Solution (Zymo Research, Irvine, CA, USA), using a Bead Mill 4 (Thermo Fisher Scientific, Waltham, MA, USA). The DNA was isolated, using ZymoBIOMICS DNA Mini Kit (Zymo Research) according to the manufacturer’s protocol. The isolated DNA was quantified, using a Qubit DNA HS Assay kit (Thermo Fisher Scientific). To isolate DNA from the rinse water, the bacteria in the water was collected using a 0.2 µm filter, and the DNA was isolated from the crashed filter (5 m/s, 300 s, using Beat Mill 4) using the ZymoBIOMICS DNA Mini Kit. 16S rRNA sequencing was performed using an Oxford Nanopore MinION (Oxford Nanopore Technologies, Oxford, UK). Full-length 16S rRNA genes were amplified with 16S_barcode_27f primer (5′-TTT CTG TTG GTG CTG ATA TTG CAG RGT TTG ATC MTG GCT CAG-3′) and 16S_barcode_1492r primer (5′-ACT TGC CTG TCG CTC TAT CTT CGG YTA CCT TGT TAC GAC TT-3′). A DNA library was constructed with PCR Barcoding Kit and Ligation Sequencing Kit (Oxford Nanopore Technologies), according to the manufacturer instruction. The DNA library was applied into MinION, and the output data were analyzed by using EPI2ME software (Oxford Nanopore Technologies).

### 3.7. Zebrafish Embryo Toxicity Test

Down and feather (0.2 g) samples were immersed in 10 mL sterile water and shaken for 1 h at room temperature. The immersion water was then collected and filtered through a 40 µm filter. Blastula stage embryos (about 5 h post fertilization (hpf)) were exposed to the immersion water and incubated at 28 °C until 72 hpf. Each group contained 10 fish with 3 triplicates. The rate of survival and number of anomalies (include hatching numbers) of embryos were recorded and imaged daily where for the AB-strain, a streomicroscope SMZ-745T (Nikon, Tokyo, Japan) equipped with an HD Lite camera (Relyon, Tokyo, Japan) was used, while for the *Tg(mpeg:EGFP)* fish, a BZX-710 fluorescent microscope (Keyence, Tokyo, Japan) was used. The numbers of macrophages in the tail fin were counted manually.

### 3.8. Quantification of Silicon Compounds in Down and Feather Samples

Detection and quantification of silicon compounds in down and feather samples were performed by the Boken Quality Evaluation Institute (Osaka, Japan). In brief, 1 g of the samples was immersed in 40 mL of tetrahydrofuran and sonicated for 10 min. The extracts were filtrated, and the solvent was removed by evaporation. Then, the Si levels were measured using IRAffinity-1 Fourier transform infrared spectroscopy (Shimadzu, Kyoto, Japan) at 600–4000 cm^−1^. To quantify the silicon compounds in down and feather samples, 0.1 g of the samples was added to 8mL of nitric acid/2 mL of hydrofluoric acid and pyrolyzed by using the Ethos Easy microwave digestion system (Milestone, Sorisole, Italy). After cooling, add 20 mL of diluted boric acid (5 to 100) to inactivate fluorine, obtain a constant volume with water and conduct quantitative analysis by using inductively coupled plasma mass spectrometry (iCAP RQ; Thermo Fisher Scientific, San Jose, CA, USA).

To evaluate Si contamination in the wash water, an AA-7000 electrothermal atomization-atomic absorption spectrometer (Shimadzu, Kyoto, Japan) equipped with a GFA-7000A graphite furnace atomizer (Shimadzu) was used.

### 3.9. Statistics

Statistical analyses were performed using Student’s *t*-test, or one-way analysis of variance with the Bonferroni-Dunn multiple comparison procedure, depending on the number of comparison, using GraphPad Prism version 8 (GraphPad Software Inc., San Diego, CA, USA). A *p*-value less than 0.05 denoted the presence of a statistically significant difference.

## 4. Conclusions

Due to an increase in the global trade of down and feather materials, the quality of the products have been brought into question. To prevent “feather pillow allergy” and harmful chemicals, we developed a quick and reliable methodology to detect biological toxicities of down and feather materials. Using this method, we can detect the potential toxicity of not only down and feather materials but also of other animal-derived products, such as leather and wool.

## Figures and Tables

**Figure 1 ijms-20-01434-f001:**
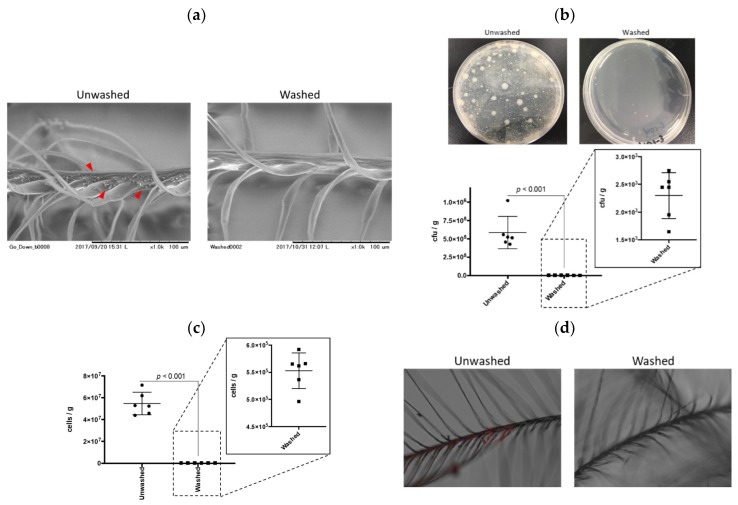
qPCR quantification of contaminated bacteria in down and feather samples. (**a**) Scanning electron microscopic images of unwashed and washed down and feather. Red arrowheads indicate debris. (**b**) Colony-forming unit (CFU) of down and feather samples. CFU was calculated using standard agar plate protocol. *n* = 6, error bars indicate SD. (**c**) qPCR analysis of bacterial numbers in down and feather samples. *n* = 6, error bars indicate SD. (**d**) Bacterial proliferation in the down and feather samples. Red signal indicates mCherry-expressing *E. coli*.

**Figure 2 ijms-20-01434-f002:**
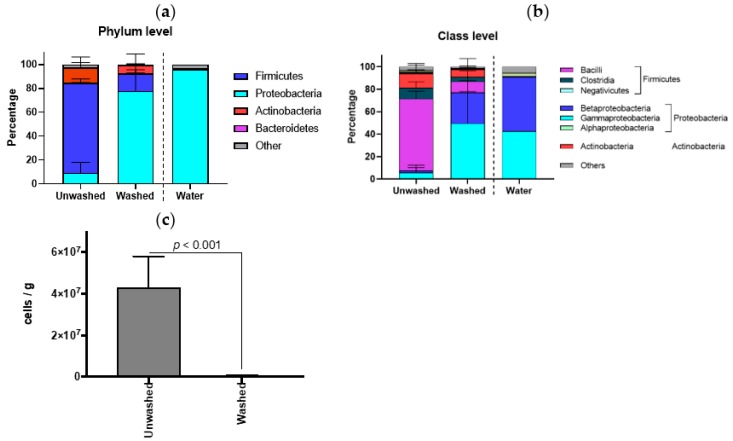
Compositions of bacteria in contaminated down and feather samples. Phylum (**a**) and class (**b**) level of the bacterial composition. *n* = 3, error bars indicate SD. Bacterial composition of each sample is depicted in Appendix A. (**c**) Quantification of contaminated bacterial in unwashed and washed samples. *n* = 3, error bars indicate SD.

**Figure 3 ijms-20-01434-f003:**
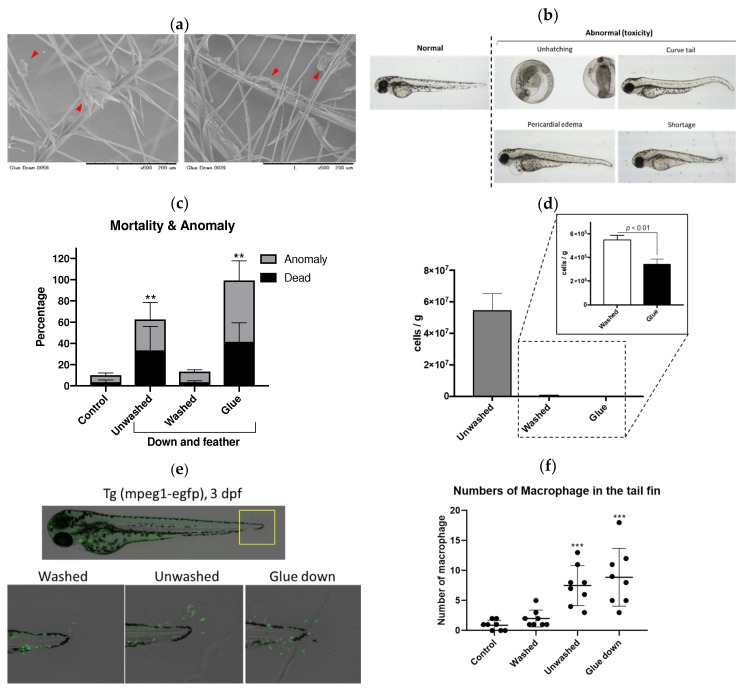
Zebrafish embryo toxicity test (ZFET) of down and feather samples. (**a**) Scanning electron microscopic images of glue down. Red arrowheads indicate glue chemicals. (**b**) Representative images of 72 hours-post-fertilization (hpf) zebrafish in ZFET. (**c**) Survivals and anomalies in ZFET. ** *p* < 0.01 vs. control. *n* = 4–6, error bars indicate SD. (**d**) Quantification of contaminated bacterial in down and feather samples used in ZFET. *n* = 4, error bars indicate SD. (**e**) Representative images of macrophage-EGFP zebrafish used in ZFET. Green indicates EGFP-expressed macrophages. Unwashed and glue down immersion water were diluted 5 times to reduce mortality. (**f**) Quantification of numbers of macrophages in the tail fin. *** *p* < 0.001 vs. control. *n* = 8, error bars indicate SD.

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
