# Peer review of "An Integrative Evaluation Method for the Biological Safety of Down and Feather Materials"

_ijms, 2019, doi:10.3390/ijms20061434_

Reviewer 1 Report

overall a good well written paper that is clear and systematic. While i am not particularly keen on the structure of the paper i can live with it and it is suitable for publication.

Author Response

Dear reviewer #2,

Thank you for your comments. We have performed some additional experiments according to the other reviewer's comments.

Sincerely yours,

Yasuhito Shimada

Reviewer 2 Report

The manuscript describes the development of integrative evaluation methods to assess biological toxicities and bacterial contamination of down and feather materials by the combination of a zebrafish embryo toxicity testing (ZFET), qPCR and 16S rRNA sequencing.

First, authors evaluated bacterial contamination of down and feather materials by using conventional agar culture method as well as qPCR-based quantification method. The results indicated that the latter method was comparable to the conventional culture method but more sensitive and higher throughput. Additionally, authors demonstrated the actual proliferation of bacteria in down and feather samples by using E. coli expressing the mCherry fluorescent protein, which indicated that suitable washing process can remove nutrition for bacterial growth.

Next, the authors performed 16S rRNA sequencing analysis of down and feather samples from different countries in order to identify the bacterial composition of washed and unwashed samples. Albeit different sample origins, unwashed samples revealed similar bacterial composition in phylum and class levels. On the other hands, the bacterial composition of the washed samples showed similarity to that of the rise water, which were dominated by proteobacteria.

Subsequently, the authors performed a zebrafish embryo toxicity test (ZFET) to evaluate the biological toxicity of the unwashed, washed and glue down samples. While no significant differences in survival rate of zebrafish embryos between these three sample groups were detected, unwashed and glue down samples showed high toxicity including a variety of abnormal phenotypes. In addition, increased the numbers of macrophages in the tail fin of animals were observed by ZFET with macrophage-EGFP zebrafish, which indicated the immunotoxicity of unwashed and glue down sample extracts.     

The manuscript is important in term of the development of sensitive and high throughput evaluation methods for bacterial contamination and biological toxicity of down and feather materials. The story is well organized in the good ordering and the experimental data support the results and conclusions. Nonetheless, I have a few suggestions and remarks.

Minor concerns

Line 109. The authors didn’t provide any reason why E.coli was used for      the bacterial proliferation experiment although Proteobacteria is not      dominant member of bacteria especially in unwashed samples.

Figure 1b. Please increase the size of pictures because it is not clear to see the washed plate picture. Graphs in Fig 1b and 1c: It would be clear to see the CFU plots for washed samples if the authors magnify the plotted sections. (as in Fig 3b).

Line 125. Please provide the information which 16s rRNA variable region was used in this experiment.

Line 156. Please explain why bacterial contamination in glue down was lower than that of unwashed samples.

Line 206. Please check the language, “1 mL of the immersion water was collected by centrifuge”. Were the bacterial cells collected from 1mL of the immersion water used for further procedure? 

Major remarks

It would be interesting to find out what kind  of chemicals cause the toxicity to zebra fish embryos. The authors can perform very easy chemical analysis of washed water from glue (before wash) and no glue feathers (after wash) by comparison with chemical standards  on  LCMS or HPLC. LCMS/ HPLC analysis only need at much for 30 min per one  measurement and the cost won’t be expensive for only two samples.

Author Response

Dear reviewer 2,

Thank you for giving us the opportunity to clarify some points of our manuscript.

We have added some experiments according to your comments.

Sincerely yours,

Yasuhito Shimada

Round  2

Reviewer 2 Report

The added quantification of silicon compounds in the revised manuscript is an excellent evidence to support the authors' proposal about toxicant contamination in glued down. The authors have addressed my concerns, and this manuscript can move forward to publish.